# Acceptability of HIV self-testing among HIV high-risk Adolescent Girls and Young Women (AGYW) in urban settings in Uganda

Laban Muteebwa [1]*, Joanita Nangendo [2,3], Dan Muramuzi [4], Shivan Nuwasiima [2], Ivan Ahimbisibwe [2], Edson Atwine [5], Cathbert Tumusiime [6], Patience A. Muwanguzi [7], Fred C. Semitala [1,3,8]

1 Department of Medicine, School of Medicine, College of Health Sciences, Makerere University, Kampala, Uganda, 2 Clinical Epidemiology Unit, School of Medicine, College of Health Science, Makerere University, Kampala, Uganda, 3 Infectious Diseases Research Collaboration, Kampala, Uganda, 4 Department of Pharmacy, School of Health Sciences, College of Health Sciences, Makerere University, Kampala, Uganda, 5 Faculty of Graduate Studies, Makerere University Business School, Makerere University, Kampala, Uganda, 6 Directorate of Research, Baylor College of Medicine Children's Foundation-Uganda, Kampala, Uganda, 7 Department of Nursing, School of Health Sciences, College of Health Sciences, Makerere University, Kampala, Uganda, 8 Makerere University Joint AIDS Program, Kampala, Uganda

* mutlabans@gmail.com

## Abstract

Adolescent Girls and Young Women (AGYW) bear a disproportionate burden of HIV in Sub-Saharan Africa, but HIV testing in this sub-population is sub-optimal. HIV self-testing (HIVST) is an evidence-based intervention that has been shown to improve testing rates among AGYW; however, its acceptability in this sub-population remains under-explored. We assessed the acceptability of HIVST and associated factors among HIV high-risk AGYW. We enrolled HIV high-risk AGYW consecutively from Kampala and Wakiso districts in a community-based cross-sectional study. A structured questionnaire was used to collect data. Acceptability was defined as willingness to use HIVST services if provided, and was measured as a composite outcome using the seven constructs of the Theoretical Framework of Acceptability (TFA) where each construct was assessed using one 5-level Likert item question weighted 1–5. The sum of scores was computed and participants with a score >21 was regarded as accepting HIVST. Descriptive statistics were used to summarize the data and a mixed effects modified Poisson was used to assess factors associated with acceptability of HIVST. Between December 2024 and May 2025, 377 HIV high-risk AGYW of mean age 19.7 years (standard deviation (SD): 2.7) were enrolled in the study of which 55.2% reported having a sexually transmitted infection in the past six months. The acceptability of HIVST was 90.2% (95%CI: 86.7, 92.8). Adjusted analysis showed that having multiple sexual partners, consistent condom use, no history of sexual violence and PrEP use in the past six months, and HIV testing in the past 12 months were significantly associated with greater willingness to use HIVST.

**Data availability statement:** The data underlying the study findings can be accessed through the public repository under DOI: https://doi.org/10.5061/dryad.95x69p8xs.

**Funding:** The findings and data reported in this publication were supported by the Fogarty International Center and the National Institute of Mental Health of the National Institutes of Health under Award Number D43 TW010037 (to FCS). The contents are solely the responsibility of the authors and do not necessarily represent the official views of the National Institutes of Health. The funders had no role in the study conceptualization, design, data collection and analysis, decision to publish or preparation of the Manuscript.

**Competing interests:** The authors have declared that no competing interests exist.

HIVST is highly acceptable among the AGYW studied, and this presents a potential opportunity that can be leveraged to scale-up HIVST services. Future implementation efforts could consider strategies that address both behavioral risks and psychological vulnerabilities to optimize uptake and impact of HIVST.

## Introduction

Globally, about 4,000 young women aged 15–24 years acquire HIV each week, with nearly two- thirds residing in Sub-Saharan Africa (SSA) [1]. In SSA, women and girls accounted for 62% of new HIV infections in 2021 [1], while in Uganda, four in five new HIV infections registered among young people 15–24 years occur in females [2]. HIV testing is the first critical entry point into HIV prevention and treatment. In 2020, the Joint United Nations Program on HIV/AIDS (UNAIDS) set the ambitious 95-95-95 target to be achieved by 2025; 95% of all people living with HIV (PLHIV) should know their status, 95% of those should receive sustained antiretroviral therapy (ART), and 95% of those on ART should achieve viral suppression [3]. However, Uganda remains behind these targets, with only 92% of PLHIV aware of their status [4], and young people representing the largest proportion of those undiagnosed. The Ministry of Health estimates that among PLHIV who do not know their status,56% are females, of whom 16% are aged 10–19 years and 48% are aged 20–34 years [5].

HIV self-testing (HIVST) is an approach where individuals collect their own specimen (oral fluid or blood), perform the test and interpret results, either alone or with someone they trust [6], which offers a promising alternative to test for HIV. HIVST is an evidence-based intervention (EBI) that has been shown to increase HIV testing uptake among "hard-to-reach," HIV high-risk populations including young people, female sex workers, and men who have sex with men [7–12]. Since 2016, HIVST has been recommended by both the World Health Organization (WHO) [6], and the Uganda Ministry of Health [13].

Despite their heightened risk of HIV acquisition, young people continue to have disproportionately low testing rates [14], driven by their unique developmental needs and a complex interplay of interpersonal, cultural, structural, and health system barriers [15]. HIVST can address many of these barriers by offering a confidential, convenient and empowering testing option [16]. However, the acceptability of this EBI among young people particularly AGYW, remains insufficiently understood and is crucial to inform effective implementation strategies. In Uganda, HIVST scale-up among AGYW has been constrained by limited awareness, restricted access, and suboptimal delivery models [17,18]. While some studies have explored HIVST acceptability among young people, most have focused on institutional settings such as universities [17,19–21], leaving a gap in understanding community-wide acceptability by AGYW. Moreover, evidence suggests that keeping young girls and women in institutional settings like schools reduces their risk of acquiring HIV [22], yet AGYW in community settings often face additional challenges, including intimate partner violence, restrictive social norms, and lack of privacy, which may further shape their willingness to use HIVST.

In this study, we assessed the acceptability of HIVST and the associated factors among HIV high-risk AGYW living in urban communities in Uganda, using the Theoretical Framework of Acceptability (TFA).

## Methods

### Study design and setting

We conducted a community-based cross-sectional study among AGYW living in the Kampala Metropolitan Area from 1st December 2024–31st May 2025. Kampala Metropolitan Area included the four most populous districts in Central Uganda including Kampala, Wakiso, Mukono and Mpigi. This is the most urbanized and commercial area in Uganda and includes its Capital City, Kampala, hosting an estimated daytime population of over five million people. The 2023 population projection estimate that there are about 688,980 AGYW in Kampala Metropolitan Areas, which is the highest number among other urban areas in Uganda [23]. The HIV prevalence among adults aged 15–49 years (2023) in Kampala, Wakiso, Mpigi and Mukono is 7.4%, 7.2%, 8.2% and 5.3% respectively, all above the national prevalence (2023) of 5.1% [4]. This region also accounted for the largest number of new HIV cases of which Wakiso had the highest (4,900) followed by Kampala (2,800), Mukono (700) and Mpigi (405) [4], and AGYW were mostly affected.

### Study population, inclusion and exclusion criteria

The study population included AGYW aged 15–24 years who had lived in the study area for at least six months, with a self-reported HIV negative status or unknown HIV status, an HIV risk score of ≥2 (high-risk) and gave written informed consent to participate in the study. The HIV risk was assessed using the HIV risk assessment tool previously used among AGYW in Kampala, Uganda [24] and follows the criteria set by the Uganda Ministry of Health. Participants who couldn't read or comprehend basic English or Luganda were excluded from the study.

### Sampling procedure

We used a multi-stage sampling design where in the first stage, two (Kampala and Wakiso) out of four districts were selected purposively because they had the highest new HIV infections in 2023 and being the most urbanized districts. At the second stage two out of four divisions in Kampala (Makindye and Rubaga) and two out of four municipalities (Entebbe and Nansana) were selected randomly. At the third stage, consecutive sampling technique was used to select the study participants. The number of participants to be selected was proportionately allocated to the study municipalities and divisions basing on the 2023 population projections of AGYW in the study areas [23].

### Data collection tools and procedure

The prospective participants were mobilized to selected sites (that provided privacy like health facilities, youth corners, drop-in centers and community halls) close to their places of residence by the Village Health Teams (VHTs). In Uganda, VHTs are community selected, volunteer community members that are trained to link communities to formal health systems. They are trained to provide health education (including in HIV and malaria), basic care, referrals, hygiene promotion and community mobilization. Each of the study areas (municipality or division) had at least two VHTs who were trained to carry out targeted community mobilization for the study. The Information about HIVST including why they were introduced, benefits, challenges and practical demonstrations on how HIVST kits are used were given to the prospective participants. Both oral (OraQuick, OraSure Technologies, Inc. Bethlehem, PA, USA) and blood-based (GOLDEN TIME, Gaobeidian PRISES Biotechnology Co., Ltd, Hebei Province, China) HIVST kits were used in the demonstration. Thereafter, eligible participants had a pre-tested structured questionnaire administered to them by trained research assistants. The questionnaire was pre-tested among 30 AGYW and adjustments made prior to data collection. The data entry screens were designed in the Kobo collect toolkit (https://www.kobotoolbox.org/) and used for data management and later the excel format of the data was exported for data cleaning in STATA version 17.0 (Texas, USA).

## Study variables

**Dependent variable.** The primary outcome was the acceptability of HIVST which was defined as the participants' willingness to use HIVST services if they were provided to them. This was measured using the seven constructs of the Theoretical Framework of Acceptability (TFA) which included; affective attitude, burden, self-efficacy, ethicality, opportunity cost, perceive effectiveness, and coherence [25]. Each construct was assessed using one 5-level Likert item question weighted from 1 to 5 (S1 File). The scores for each question were summated to give a total score for which the high score represents high willingness to use HIVST services. The total score was then dichotomized using the 50th percentile (21) of the possible sum of scores that range from 7-35, so that those with a score >21 are regarded to "accept the use of HIVST" and those ≤21 don't accept the use of HIVST, an approach that has been used prior [26,27].

**Independent variables.** The independent variables included; socio-demographics like age in completed years, religion, marital status (live with primary partner or don't), schooling status, employment status, completed education level, and monthly income (in United States Dollar); behavioral factors like sexual practices in past six months (multiple sexual partners, sex under influence of recreational drugs, sex for money), ever been pregnant, previous use of Post Exposure Prophylaxis (PEP), or Pre-Exposure Prophylaxis (PrEP) and having a sexually transmitted infection in the past six months defined as having had a diagnosis or an abnormal vaginal discharge; and HIV testing practices like being ever tested, testing in past three months, testing in past 12 months, knowledge about HIVST, source of knowledge, ever used HIVST kit, and ever looked for HIVST kit). Partner characteristics included partner's age, and occurrence of intimate partner violence (IPV) and/or occurrence of sexual violence from a sexual partner in the past six months.

## Data analysis

Data analysis was conducted in STATA version 17.0 (Texas, USA). The continuous variables were summarized using mean and standard deviation (SD) or the median and interquartile range (IQR). The categorical variables were summarized using frequencies and percentages. The acceptability scores were summarized using median, interquartile range and the range. The proportion of participants who were willing to use HIVST was computed and the logit 95% confidence interval (CI) adjusted for multistage sampling design was computed. The factors associated with acceptability of HIVST were assessed using a mixed effects modified Poisson regression model with robust standard errors adjusted for the multi-stage sampling design. At unadjusted analysis the crude prevalence ratios (cPR), their 95% CIs and P-values were computed. The independent variables with a P-value of ≤0.2 were considered for the adjusted analysis. At the adjusted analysis the model was built following a backward elimination approach, interaction was assessed using a chunk test and independent variables that change the aPR of another by a magnitude of 10% were retained in the final model. The interaction was assessed among variables that stayed in the basic model including consistent condom use, being tested in past 12 months, having >1 sexual partner, using PrEP and having experiences sexual violence in the past six months. For the final model, the level of significance was set at 0.05 and the adjusted prevalence ratios (aPR), their corresponding 95% CIs and P-values were computed.

**Ethics statement.** The study was reviewed and approved by the Makerere University School of Medicine Research Ethics Committee (Mak-SOMREC) under the reference number Mak-SOMREC-2023–843. Further clearance was sought from the Uganda National Council for Science and Technology (UNCST) under the reference number HS5167ES. The administrative clearances to conduct the study were obtained from the Kampala Capital City Authority (KCCA), and the Municipal Health Offices of Entebbe and Nansana Municipalities. The adult and emancipated minors provided written informed consent, while the other minors (<18 years) gave written informed assent in addition to their guardian's written informed consent following the UNCST guidelines [28].

## Results

### Socio-demographic characteristics of study participants

We enrolled 377 HIV high-risk AGYW with mean (SD) age of 19.7 (2.7) years. About two-thirds (69.0%) were from Wakiso district and (65.5%) were not in school. Majority of the participants were Christians (70.8%), and 46.4% had completed secondary level education. Slightly more than half of the participants (58.4%) reported to be employed, and the median monthly income was United States Dollars (USD) $41.6 (IQR: 27.7 – 55.5) (Table 1).

### Behavioral and partner characteristics of study participants

About half of the participants reported to have had a sexually transmitted infection (STI) (55.2%) in the past six months, nearly a third (30.0%) reported to have had sex in exchange for money or gifts, and 11.7% reported to have had anal sex in the past 6 months. In the 6 months preceding data collection, only 13.3% of participants reported consistent condom use, while 6.6% reported using post exposure prophylaxis (PEP) and just 5.3% reported to have used pre-exposure prophylaxis (PrEP). Majority of the participants had ever had an HIV test (82.8%) of which close to three-quarters (73.7%) had an HIV test in the past 12 months and just close to half had tested in the past 3 months (47.8%). Most of the participants had the most recent HIV test at the health facility (71.7%), and about 15.2% had an HIV test from their homes. Whereas 70.0% of the participants had heard about HIVST, just about half (49.6%) had ever used an HIVST and about two thirds (62.1%) reported to have never tried to look for an HIVST kit (Table 2).

### Acceptability of HIVST among AGYW

The median acceptability score was 31 (IQR: 28, 34) with a range of 8–35. The acceptability of HIVST was 90.2% (95% CI: 86.7, 92.8).

**Table 1. Demographic characteristics of 377 AGYW (15-24 years) recruited from Kampala and Wakiso districts.**

| Variable | Categories | Frequency (N = 377) | Percentage (%) |
|---|---|---|---|
| Age (years) | Mean (SD) | 19.7 (2.7) | |
| Study districts | Kampala | 117 | 31.0 |
| | Wakiso | 260 | 69.0 |
| Live with primary partner | Yes | 103 | 27.3 |
| | No | 274 | 72.7 |
| Religion | Christian | 267 | 70.8 |
| | Moslem | 107 | 28.3 |
| | None | 3 | 0.8 |
| Schooling status | In school | 130 | 34.5 |
| | Not in school | 247 | 65.5 |
| Education level completed | None | 18 | 4.8 |
| | Primary level | 166 | 44.0 |
| | Secondary level | 175 | 46.4 |
| | Tertiary level | 18 | 4.8 |
| Employment status | Employed | 157 | 41.6 |
| | Not employed | 220 | 58.4 |
| Monthly income (USD)* | Median (IQR) | $41.6 (IQR: 27.7 – 55.5) | |

*USD – United States Dollar (Average rate: 1 Uganda shillings= $0.00028).

**Table 2. Behavioral and partner characteristics of 377 AGYW (15-24 years) recruited from Kampala and Wakiso districts.**

| Variable | Categories | Frequency (N = 377) | Percentage (%) |
|---|---|---|---|
| Had anal sex in past 6 months | Yes | 44 | 11.7 |
| | No | 333 | 88.3 |
| Had sex for money in past 6 months | Yes | 113 | 30.0 |
| | No | 264 | 70.0 |
| Had an STI in past six months@ | Yes | 208 | 55.2 |
| | No | 169 | 44.8 |
| Consistent condom use in the past 6 months | Yes | 50 | 13.3 |
| | No | 327 | 86.7 |
| Used recreational drug before or during sex in past 6 months | Yes | 58 | 15.4 |
| | No | 318 | 84.6 |
| Used PEP in past 6 months$ | Yes | 25 | 6.6 |
| | No | 352 | 93.4 |
| Used PrEP in past 6 months# | Yes | 20 | 5.3 |
| | No | 357 | 94.7 |
| Ever tested for HIV | Yes | 312 | 82.8 |
| | No | 65 | 17.2 |
| Tested in past 12 months* | Yes | 230 | 73.7 |
| | No | 82 | 26.3 |
| Tested in past 3 months* | Yes | 149 | 47.8 |
| | No | 163 | 52.2 |
| Point of most recent HIV test* | Health facility | 165 | 71.7 |
| | Community outreach | 28 | 12.2 |
| | At Home | 35 | 15.2 |
| | Others | 2 | 0.9 |
| Heard of HIVST | Yes | 260 | 70.0 |
| | No | 117 | 30.0 |
| Ever used HIVST** | Yes | 129 | 49.6 |
| | No | 131 | 50.4 |
| Ever looked for HIVST kit in the neighborhood and failed | Yes | 42 | 11.1 |
| | No | 101 | 26.8 |
| | Never tried | 234 | 62.1 |
| Primary partner older by >5 years*** | Yes | 155 | 50.3 |
| | No | 153 | 49.7 |
| Faced sexual violence in past 6 months | Yes | 45 | 11.9 |
| | No | 332 | 88.1 |

@STI-Sexually Transmitted Infection, $PEP-Post Exposure Prophylaxis, #PrEP-Pre-Exposure Prophylaxis, *N = 312, **N = 260, ***N = 308.

## Factors associated with acceptability of HIVST

At adjusted analysis, consistent use of condoms in past six months (aPR: 1.10 (95% CI: 1.05, 1.15)), having multiple sexual partners (aPR: 1.08 (95% CI: 1.03, 1.13)) and not having a history of sexual violence in past six months (aPR: 1.20 (95% CI: 1.15, 1.25)) were significantly associated with acceptability of HIVST among HIV high–risk AGYW. Additionally, the participants who had not used PrEP in the past six months were more likely to accept HIVST (aPR: 1.11 (95% CI:

1.10, 1.12)), while those who had had an HIV test in the past 12 months were more willing to use HIVST (aPR: 1.12 (95% CI: 1.00, 1.26)) (Table 3).

## Discussion

In this study, we assessed the acceptability of HIVST and associated factors among HIV high-risk AGYW in Kampala Metropolitan Area. Acceptability of HIVST among HIV high-risk AGYW was very high. Having multiple sexual partners, consistent use of condoms, use of PrEP in the past six months, as well as having an HIV test in past 12 months were positively associated with willingness to use HIVST. Whereas having a history of sexual violence in the past six months was negatively associated with willingness to use HIVST. The high acceptability of HIVST can therefore be leveraged to scale-up this EBI among HIV high-risk AGYW. However, such scale-up efforts must appreciate both the behavioral risk factors and psychological vulnerabilities of AGYW in order to target HIVST intervention appropriately to optimize uptake.

In this study, we found out that nine out of every ten HIV high-risk AGYW were willing to use HIVST services if they were provided to them, despite low actual uptake where only about half had ever used an HIV self-test. This disparity could be because HIVST is still a relatively new approach of HIV testing in Uganda, and is currently in few public health facilities, and some private health facilities and pharmacies. Nonetheless, a recent scoping review reported issues related to cost of the kits and limited post-test counselling as important barriers to uptake of HIVST services [29]. Other similar community-based studies have reported a high levels of acceptability of HIVST among adolescents and adults in Zambia [30]. Whereas the acceptability of HIVST was high in this community-wide sample of high risk AGYW, similar findings were reported in studies conducted among young people in academic institutional settings in Uganda and South Africa where the acceptability was 93% and 87.1% respectively [17,31]. However, some other studies done in low-income countries have reported a lower acceptability of HIVST among young people in institutional settings [32]. The differences could be due to the HIV risk profile and perceived risk between young women in institutional settings and those that live in the general community. The high acceptability of HIVST in our study presents an opportunity for policy makers and HIV programmers to scale-up HIVST intervention while optimizing its unique attributes to reach HIV high risk AGYW living in urban settings and improve their HIV testing practices.

In this study, HIV high-risk AGYW who reported to have had an HIV test in the past 12 months were more likely to accept the use of HIVST. In studies conducted among young university students in Uganda [17] and Nigeria [21], recent prior HIV testing experience was similarly associated with acceptability of HIVST. HIV testing has been associated with heightened anxiety and/or fear [33] stemming from the testing process and the test outcome, much as prior testing experience can help in reducing them at subsequent HIV testing. In the conventional HIV testing approaches, pre-test counselling delivered by health workers can help reduce anxiety related to HIV testing [34]. HIVST especially where an individual is unassisted, one may have heightened anxiety which may impede the use of HIVST services. Nonetheless, HIVST may be particularly attractive to high-risk AGYW because it not only reduces the burden of repeated facility visits for routine HIV testing and the associated long waiting times [35], but also it may offer greater control over the testing process. However, it is important for HIV programmers and policymakers to explore additional strategies to reach the first-time testers and those with limited contact with health services, as they may be less likely to accept HIVST without targeted education and support.

HIV high-risk AGYW that had multiple sexual partners (more than one sexual partner) were 1.08 times more likely to accept HIVST compared to those who had only one stable sexual partner. Having multiple sexual partners increases the risk perception among AGYW which may drive the desire for routine HIV testing. Whereas conventional HIV testing is seen as inconvenient, stigmatizing, or often inaccessible, HIVST offers a discreet and private alternative to AGYW because it can be done at a convenient time and location. Moreover, HIVST allows testing before or after sexual encounters with different partners without requiring repeated visits to the health facility and this may drive acceptability and eventually adoption of HIVST. A recent study conducted among female university students in Uganda reported similar findings

**Table 3. Unadjusted and adjusted analysis of factors associated with willingness to use HIVST services.**

| Variables | Acceptability of HIVST | Unadjusted analysis | | Adjusted analysis | |
|---|---|---|---|---|---|
| | n = 340 (%) | cPR (95% CI) | P value | aPR (95% CI) | P-value |
| Age (years) | | | | | |
| 15 – 19 years | 160(89.4) | Ref | | | |
| 20 – 24 years | 180 (90.9) | 1.02 (0.92, 1.13) | 0.750 | | |
| Live with primary partner | | | | | |
| Yes | 96 (93.2) | 1.05 (0.97, 1.13) | 0.240 | | |
| No | 244 (89.1) | Ref | | | |
| Schooling status | | | | | |
| In school | 116 (89.2) | 0.98 (0.84, 1.15) | 0.838 | | |
| Not in school | 244 (90.7) | Ref | | | |
| Consistent condom use in the past 6 month | | | | | |
| Yes | 48 (96.0) | 1.08 (1.07, 1.08) | <0.001 | 1.10 (1.05, 1.15) | <0.001 |
| No | 292 (89.3) | Ref | | Ref | |
| Had an STI in past six months | | | | | |
| Yes | 189 (90.9) | 1.02 (1.01, 1.02) | <0.001 | | |
| No | 151 (89.4) | Ref | | | |
| Shared injections | | | | | |
| Yes | 15 (88.2) | 0.98 (0.89, 1.07) | 0.618 | | |
| No | 325 (90.3) | Ref | | | |
| Ever tested for HIV | | | | | |
| Ever tested | 283 (90.7) | 1.03 (0.93, 1.15) | 0.520 | | |
| Never tested | 57 (87.7) | Ref | | | |
| Tested in past 12 months* | | | | | |
| Yes | 214 (93.0) | 1.11 (1.02, 1.20) | 0.015 | 1.12 (1.00, 1.26) | 0.047 |
| No | 69 (84.2) | Ref | | Ref | |
| Ever heard about HIVST | | | | | |
| Yes | 242 (93.1) | 1.11 (0.95, 1.30) | 0.191 | | |
| No | 98 (83.8) | Ref | | | |
| Ever used HIVST | | | | | |
| Yes | 122 (94.6) | Ref | | | |
| No | 120 (91.6) | 0.97 (0.92, 1.03) | 0.275 | | |
| Ever been pregnant | | | | | |
| Yes | 147 (90.2) | 1.00 (0.87, 1.16) | 1.000 | | |
| No | 193 (90.2) | Ref | | | |
| Current Contraceptive use | | | | | |
| Yes | 190 (92.7) | 1.06 (0.92, 1.23) | 0.417 | | |
| No | 150 (87.2) | Ref | | | |
| Had > 1 sexual partner in past six months | | | | | |
| Yes | 103 (93.6) | 1.06 (0.98, 1.14) | 0.186 | 1.08 (1.03, 1.13) | 0.001 |
| No | 237 (88.8) | Ref | | Ref | |
| Had sex in exchange for money in past six months | | | | | |
| Yes | 101 (89.4) | 0.99 (0.82, 1.19) | 0.893 | | |
| No | 239 (90.5) | Ref | | | |
| Used PrEP in past six months# | | | | | |
| Yes | 16 (80.0) | Ref | | Ref | |
| No | 324 (90.8) | 1.13 (1.13, 1.14) | <0.001 | 1.11 (1.10, 1.12) | <0.001 |

*(Continued)*

**Table 3.** (Continued)

| Variables | Acceptability of HIVST | Unadjusted analysis | | Adjusted analysis | |
|---|---|---|---|---|---|
| | n = 340 (%) | cPR (95% CI) | P value | aPR (95% CI) | P-value |
| Used recreational drugs before or during sexual intercourse | | | | | |
| Yes | 53 (91.4) | 1.02 (0.82, 1.27) | 0.890 | | |
| No | 287 (90.0) | Ref | | | |
| Faced sexual violence in past 6 months | | | | | |
| Yes | 36 (80.0) | Ref | | Ref | |
| No | 304 (91.0) | 1.15 (1.10, 1.19) | <0.001 | 1.20 (1.15, 1.25) | <0.001 |
| Primary partner older by >5 years** | | | | | |
| No | 142 (92.8) | 1.04 (1.02, 1.05) | <0.001 | | |
| Yes | 139 (89.7) | Ref | | | |

#PrEP-Pre-Exposure Prophylaxis, *N=312, **N=308.

[17]. Whereas in a study among 496 adolescents conducted in Mozambique, over 80% of participants chose HIVST over conventional testing in a health facility [36]. On the other hand, a cross sectional study conducted among trans-gender women in Malaysia, those who had multiple sex clients per day were less likely to be willing to accept HIVST [37]. This contradicting finding could be because trans-gender women face far more discrimination and stigma related to their sexual orientation and the most often transactional nature of their sexual relationships like in the case of the Malaysian study. This finding demonstrates that HIVST has potential to increase reach of HIV testing services to AGYW at high-risk, particularly those with multiple sexual partners. Therefore, HIV prevention programs should prioritize making HIVST services accessible to high-risk AGYW since they are more likely to be adopted.

In our study, HIV high-risk AGYW who had not experienced sexual violence from any of their sexual partners were 1.2 times more likely to accept HIVST compared to those who had faced sexual violence in the past six months preceding data collection. Sexual violence can lead to both physical and psychological trauma, including anxiety, depression and fear of further medical interactions including HIV testing. For example, in a cohort study conducted in urban Kisumu, Kenya among women aged 18–39 years who were given HIV test kits to test with their partners, those who had a recent history of intimate partner violence (IPV) were less likely to utilize the test kits [38]. In another cohort study conducted among 265 female sex workers in Malawi who had HIV self-test kits delivered to them, just about 13% of those who reported sexual abuse used the test kits [39]. AGYW who experience sexual violence may also associate HIV testing with such a traumatic event, making it a distressing experience [40]. In fact, in a cross sectional study conducted among 271 medical students in Tanzania, participants reported IPV as one of the reasons for non-uptake of HIVST [41]. AGYW without recent sexual violence history may feel safer, more in control of their bodies and health decisions, and thus more confident and willing to use HIVST. This finding underscores the need for trauma-informed approaches when promoting HIVST, and scale-up efforts should recognize that recent survivors of sexual violence may need additional support.

AGYW who reported consistent condom use in the past 6 months before the study, were more likely to accept HIVST compared to those who didn't use condoms or those who used them inconsistently. Consistent condom users often demonstrate a high level of sexual health awareness and risk perception, and thus more open to HIVST that empowers them to monitor their health privately and regularly. Previous studies conducted among university students [21] and transgender women [37] have similarly demonstrated that individuals who consistently use condoms are more likely to accept HIVST as an additional tool to maintain their sexual health. Such behavior reflects an individual's intentional effort to reduce HIV and STI transmission risk by being proactive and thus being more receptive to HIVST. Moreover, HIVST appeals to such individuals because it offers a private, convenient, and fast way to test without needing to go to a health

facility. This finding further demonstrates that integrating HIVST into the broader prevention efforts like condom delivery platforms may improve its adoption and HIV testing rate. Nonetheless, this finding demonstrates that high-risk AGYW with a low-risk perception evidenced by no or inconsistent condom use may need additional support to use HIVST.

Additionally, HIV high-risk AGYW who reported to have used PrEP for HIV prevention in the past six months were less likely to accept HIVST compared to PrEP non-users. Routine PrEP users are often already connected to regular health services (including PrEP refills/injections) where routine HIV counselling and testing is typically provided, thus may not feel the need for additional HIVST. However, a previous qualitative study conducted in Kenya among AGYW who had and hadn't used PrEP reported that both groups were highly willing to use HIVST if provided [42]. Whereas non-PrEP users may not be routinely engaged in HIV prevention services, yet they still want to monitor their HIV status, and thus HIVST offers a low-barrier method to test. In this case, HIVST fills a service gap for those not receiving PrEP but who are still at an elevated risk of HIV acquisition. This finding underscores the need for integrated, differentiated HIV prevention strategies that offer both PrEP and HIVST tailored to individuals' preferences as recently recommended by WHO [43].

## Limitations

The study had various limitations. Firstly, the data on some of the variables were collected retrospectively and there-fore, prone to recall bias. However, this was mitigated by limiting the period of recall to utmost six (6) months except for prior HIV testing which was stretched to twelve (12) months. Secondly, we consecutively sampled participants from the selected municipalities and divisions so could have introduced selection bias and limits generalizability. Thirdly, the pri-mary outcome of the study (acceptability of HIVST) was measured using the theoretical framework of acceptability (TFA) that has been used previously in other studies to measure acceptability of health interventions among AGYW in Uganda [26,44]. The TFA has not yet been validated for use in our study setting and therefore, may affect the validity of the results. However, the TFA was very reliable in measuring acceptability of HIVST services with a reliability coefficient of 0.89. Fourthly, we excluded participants who didn't read or write English or Luganda, and this may have contributed to selection bias, however, the local language Luganda is widely used in our study setting even by those with low levels of education. In our study we assessed the perceived willingness to use HIVST (acceptability) which may have significant differences with actual uptake of the testing service. Additionally, we categorized AGYW who self-reported to have had an abnormal vaginal discharge in the past six months as having an STI and this may have introduced misclassification bias. However, given that we studied sexually active and HIV high-risk AGYW, history of an STI can be an important proxy for having an STI. Nevertheless, we believe that the systemic biases mentioned above could have had minimal impact on the study findings.

## Conclusion

HIVST is highly acceptable among HIV high-risk AGYW studied, and this presents a potential opportunity that can be lev-eraged to scale-up HIVST services. Future implementation efforts could consider strategies that address both behavioral risks and psychological vulnerabilities to optimize uptake and impact of HIVST.

## Recommendations

Policy makers should capitalize on the high acceptability of HIVST by scaling-up HIVST services and carrying out commu-nity awareness campaigns targeting both in-school and out-of-school high-risk AGYW.

Program implementers should design and deliver targeted, context-sensitive HIVST interventions that address behav-ioral and psychological barriers to uptake.

Researchers should develop and evaluate age- and gender-responsive HIVST delivery models to further enhance uptake and impact among high-risk AGYW.

## Supporting information

**S1 File. Questionnaire – A questionnaire that was used in this study to capture participant responses.**
(PDF)

## Acknowledgments

We would like to acknowledge all the amazing participants that gave in their time to participate in the study. We also acknowledge the Village Health Teams of Kampala and Wakiso districts for supporting us in mobilization of the study participants.

## Author contributions

**Conceptualization:** Laban Muteebwa, Joanita Nangendo, Dan Muramuzi, Ivan Ahimbisibwe, Edson Atwine, Cathbert Tumusiime, Patience A. Muwanguzi, Fred C. Semitala.

**Data curation:** Cathbert Tumusiime.

**Formal analysis:** Laban Muteebwa, Dan Muramuzi, Shivan Nuwasiima, Ivan Ahimbisibwe, Cathbert Tumusiime.

**Funding acquisition:** Fred C. Semitala.

**Methodology:** Laban Muteebwa, Joanita Nangendo, Dan Muramuzi, Shivan Nuwasiima, Ivan Ahimbisibwe, Edson Atwine, Cathbert Tumusiime, Patience A. Muwanguzi, Fred C. Semitala.

**Project administration:** Laban Muteebwa, Joanita Nangendo, Edson Atwine, Patience A. Muwanguzi, Fred C. Semitala.

**Supervision:** Laban Muteebwa, Joanita Nangendo, Patience A. Muwanguzi, Fred C. Semitala.

**Validation:** Laban Muteebwa, Shivan Nuwasiima, Edson Atwine, Cathbert Tumusiime, Patience A. Muwanguzi, Fred C. Semitala.

**Writing – original draft:** Laban Muteebwa, Dan Muramuzi, Shivan Nuwasiima, Edson Atwine.

**Writing – review & editing:** Laban Muteebwa, Joanita Nangendo, Dan Muramuzi, Shivan Nuwasiima, Ivan Ahimbisibwe, Edson Atwine, Cathbert Tumusiime, Patience A. Muwanguzi, Fred C. Semitala.

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
