## [Decision Letter · Decision Letter 0]

15 Dec 2025

PGPH-D-25-01949

Acceptability of HIV self-testing among HIV high-risk Adolescent Girls and Young Women (AGYW) in urban settings in Uganda

Dear Dr. Muteebwa,

Thank you for submitting your manuscript to PLOS Global Public Health. After careful consideration, we feel that it has merit but does not fully meet PLOS Global Public Health’s publication criteria as it currently stands. Therefore, we invite you to submit a revised version of the manuscript that addresses the points raised during the review process.

We look forward to receiving your revised manuscript.

Kind regards,

Janet Seeley

Academic Editor

Journal Requirements:

1. Please include a copy of questionnaire you used to collect data as Supplementary Information in your revised manuscript.

1. Please clarify all sources of funding (financial or material support) for your study. List the grants (with grant number) or organizations (with url) that supported your study, including funding received from your institution.

2. State the initials, alongside each funding source, of each author to receive each grant.

3. State what role the funders took in the study. If the funders had no role in your study, please state: “The funders had no role in study design, data collection and analysis, decision to publish, or preparation of the manuscript.”

4. If any authors received a salary from any of your funders, please state which authors and which funders.

3. Your manuscript is missing the following sections: Introduction. Please ensure these are present, and in the correct order, and that any references to subheadings in your main text are correct. An outline of the required sections can be consulted in our submission guidelines here:

https://journals.plos.org/globalpublichealth/s/submission-guidelines#loc-parts-of-a-submission

Additional Editor Comments:

Both reviewers have highlighted areas in the text which are not clear, as well as making helpful comments on the content. I hope you find their input helpful in developing your paper.

Reviewers' comments:

Reviewer's Responses to Questions

**Comments to the Author**

1. Does this manuscript meet PLOS Global Public Health’s publication criteria?

Reviewer #1: Yes

Reviewer #2: Yes

2. Has the statistical analysis been performed appropriately and rigorously?

Reviewer #1: Yes

Reviewer #2: Yes

3. Have the authors made all data underlying the findings in their manuscript fully available (please refer to the Data Availability Statement at the start of the manuscript PDF file)?

Reviewer #1: Yes

Reviewer #2: Yes

4. Is the manuscript presented in an intelligible fashion and written in standard English?

Reviewer #1: Yes

Reviewer #2: Yes

Reviewer #1: This manuscript presents an important topic on the acceptability of HIV self-testing services. Overall, the manuscript is well organised and written clearly. I have some suggestions to improve the manuscript:

Major revisions:

1. Please expand lines 84-85 to include that you also explored which behavioural and clinical factors are associated with acceptability of HIVST.

2. Can the authors expand more on VHTs?

2.1. What is the composition (sex and age) of these VHTs?

2.2. How many teams were used in this study?

2.3. Were VHTs provided with any training? What was provided as part of training?

3. Dependent variable:

3.1. While the composite scoring and median split provide a simple way to classify participants, this method risks information loss and relies on an arbitrary cut-off. A more robust approach would be to analyse the summed TFA score as a continuous outcome, which preserves variability and statistical power. However, it seems that a binary definition of acceptability is desired for interpretability; as such, I suggest conducting sensitivity analyses with multiple thresholds (e.g., >28, >35) to show robustness.

4. Line 149: “...having a sexually transmitted disease in the past six months defined as having had a diagnosis or an abnormal vaginal discharge...” This is a bit awkward, as one is a confirmed condition, while the other is a non-specific symptom. Can you please clarify whether those with abnormal vaginal discharge were treated for STIs, i.e., syndromic management, and thus were classified as having an STI?

5. Line 210-212: Acceptability of HIVST among AGYW:

5.1. Since TFA is multidimensional, authors could also report construct-specific scores (e.g., mean Likert scores for affective attitude, burden, self-efficacy, etc.). Perhaps provide a table or a figure.

6. In table 3: Please add a column to show the number and percentages of those willing to use HIVST services for each variable/category.

7. Line 250: Given the sampling technique, which is a non-probability method, i.e., the sample was not randomly selected from the target population, generalisability might not hold. As such, I suggest rephrasing "… acceptability of HIVST in our population…" to say "in our study".

8. Lines 356-360 Conclusion:

8.1. The conclusion seems to overstate the findings relative to the study design and sample. Specifically, the statement that HIVST is “highly acceptable among HIV high-risk AGYW in urban settings” and the suggestion to scale up services imply broader generalisability and causal evidence that the study design does not support. I recommend revising the conclusion to clarify that acceptability was observed within the enrolled sample.

8.2. Scale-up recommendations should be framed cautiously, while still noting the potential value of integrating behavioural and psychological considerations, e.g., future implementation efforts could consider strategies addressing both behavioural risks and psychological vulnerabilities to optimise uptake and impact.

8.3. It would be helpful to mention that generalisability is limited by the use of consecutive sampling and the urban-specific study context and that further research is needed to confirm acceptability in other populations and settings.

Minor revisions:

1. I am not familiar with the KoboCollect toolkit. So, perhaps if wording allows, authors can mention what this tool is or provide any reference.

2. Line 149: rephrase “sexually transmitted disease” to “sexually transmitted infection”

3. Line 164-165: “At unadjusted analysis the crude prevalence ratios (cPR) were computed, their 95% CIs and P values were computed.” I suggest revising to avoid repetition: At unadjusted analysis the crude prevalence ratios (cPR) and their 95% CIs and P values were computed.

4. As a standard, please put a hyphen between “P value”, i.e., p-value.

5. Line 168: Can the authors specify which interaction terms were assessed?

6. Line 170-171: “the level of significance was set at 0.05 and the corresponding adjusted prevalence ratios (aPR), their corresponding 95% CIs and P values were computed” Similarly, there’s a repetition. I suggest removing the first “corresponding”.

7. Line 184: “We enrolled in 377…” remove “in”

8. Please ensure that all abbreviations in the tables are listed below that table.

9. I suggest restructuring lines 255–260 for improved logical flow. Specifically, the sentence “Whereas HIV testing has been associated with heightened anxiety and/or fear … help in reducing them at subsequent HIV testing” could be moved to precede the sentence starting “In the conventional HIV testing approaches, pre-test counselling delivered by health workers…”

Reviewer #2: COMMENTS

The authors sought to determine the acceptability of HIVST among HIV high-risk AGYW living in urban communities in Uganda, using the Theoretical Framework of Acceptability (TFA).

Abstract

Nil

Introduction

1. The introduction cites HIV incidence data from 2021. It is recommended to reference the most recent UNAIDS estimates to strengthen the relevance of the study in the current context.

Methods

2. Please provide more detail on how participants were identified in the community using consecutive sampling. Specifically, clarify the locations and processes used to recruit potential participants. This information is critical for assessing whether the study population can be considered representative of the broader community.

3. Please clarify how the participant numbers were proportionately allocated using the 2023 population projections of AGYW in the study areas. Was this allocation based on a ratio between the areas, or was it part of the sample size calculation methodology? Providing this detail will help readers understand the sampling approach and its rationale.

4. Clarify whether self-reported HIV status was considered in the eligibility criteria. To accurately assess the acceptability of HIV self-testing, it may be important to include only individuals who are not living with HIV. If this was not the case, explain the rationale for the chosen approach.

5. The manuscript notes that information on HIVST kits and their use was provided to prospective participants. More detail on the specific content of this information would help clarify how it may have influenced participants’ perceptions and the acceptability of the interventions.

6. The questionnaire items are not included, making it difficult to assess how the questions align with the Theoretical Framework of Acceptability (TFA). Including the questionnaire as supporting information would enhance transparency and allow readers to evaluate the connection between the items and the framework.

7. The manuscript indicates that a Likert scale was used, but the meaning of responses 1–5 is not described. Clarifying what each response represents is essential to interpret what a score greater than 21 signifies. Including the study questionnaire as supporting information would further enhance transparency.

8. The reported p-value cutoff of 0.2 for adjusted analysis appears high. Clarify how this threshold was determined and justify its use in the context of the study’s analytical approach.

Results

9. In the unadjusted analysis, “Had an STI in the past six months” and “Primary partner older by 5 years” had p-values <0.2 but were not included in the adjusted model, despite the description in the methods. Clarify the reason for excluding these variables from the adjusted analysis.

10. Table 2 lists “condom use in the past 6 months” as a yes/no item, while the text refers to consistent condom use. This discrepancy makes it unclear whether the table reflects consistent use. Clarify the terminology or update the table to ensure alignment with the description in the text.

11. In Table 2, the item “Used drug before or during sex” is ambiguous. Clarify whether this refers to recreational drugs, prescription medication, or another category to ensure accurate interpretation.

12. The results section reports a median acceptability score, but the methods do not describe how this analysis was conducted. Include details of the approach in the methods to ensure consistency.

13. Include the definition for “STI in the past 6 months” in the legend of Table 2, as described in the text. This will ensure the table can stand alone and be fully interpretable without referring to the main text.

14. Partner characteristics are not presented in Table 1 or Table 2. Including the proportions for these characteristics would provide important context for interpreting the unadjusted and adjusted models and understanding their relevance to the study population.

Discussion

15. The discussion references a recent study among female university students in Uganda reporting similar findings but does not specify what those findings were. Including the key results from that study would substantiate the comparison and strengthen the argument.

Grammar and formatting:

16. Check for consistency in spacing between words in text and in text reference.

17. In the discussion section, try to minimize the overuse of transitional words such as however, whereas, furthermore, moreover, and nonetheless. Excessive use of these connectors can make the text feel repetitive and disrupt the natural flow. Aim to structure sentences so that the logical progression is clear without relying heavily on these linking words. This will help maintain readability and improve the overall coherence of your discussion.

**Do you want your identity to be public for this peer review?** For information about this choice, including consent withdrawal, please see our Privacy Policy

Reviewer #1: No

Reviewer #2: No

---

## [Decision Letter · Decision Letter 1]

9 Feb 2026

Acceptability of HIV self-testing among HIV high-risk Adolescent Girls and Young Women (AGYW) in urban settings in Uganda

PGPH-D-25-01949R1

Dear Muteebwa,

We are pleased to inform you that your manuscript 'Acceptability of HIV self-testing among HIV high-risk Adolescent Girls and Young Women (AGYW) in urban settings in Uganda' has been provisionally accepted for publication in PLOS Global Public Health.

Best regards,

Janet Seeley

Academic Editor

Thank you for paying careful attention to the reviewers' comments.

Reviewer Comments (if any, and for reference):

Reviewer's Responses to Questions

**Comments to the Author**

Reviewer #1: All comments have been addressed

Reviewer #2: All comments have been addressed

publication criteria?

Reviewer #1: Yes

Reviewer #2: Yes

3. Has the statistical analysis been performed appropriately and rigorously?

Reviewer #1: Yes

Reviewer #2: Yes

4. Have the authors made all data underlying the findings in their manuscript fully available (please refer to the Data Availability Statement at the start of the manuscript PDF file)?

Reviewer #1: Yes

Reviewer #2: Yes

5. Is the manuscript presented in an intelligible fashion and written in standard English?

Reviewer #1: Yes

Reviewer #2: Yes

Reviewer #1: I assessed the revised manuscript by comparing it directly with the previous version and my original review. The authors’ revisions satisfactorily address my comments, and I have no additional concerns. I recommend the manuscript for publication.

Reviewer #2: (No Response)

**Do you want your identity to be public for this peer review?** For information about this choice, including consent withdrawal, please see our Privacy Policy

Reviewer #1: No

Reviewer #2: No
